# Transient Receptor Potential Canonical 5 (TRPC5): Regulation of Heart Rate and Protection against Pathological Cardiac Hypertrophy

**DOI:** 10.3390/biom14040442

**Published:** 2024-04-04

**Authors:** Pratish Thakore, James E. Clark, Aisah A. Aubdool, Dibesh Thapa, Anna Starr, Paul A. Fraser, Keith Farrell-Dillon, Elizabeth S. Fernandes, Ian McFadzean, Susan D. Brain

**Affiliations:** 1BHF Cardiovascular Centre of Research Excellence, School of Cardiovascular and Metabolic Medicine & Sciences, King’s College London, London SE1 9NH, UKjames.2.clark@kcl.ac.uk (J.E.C.); a.aubdool@qmul.ac.uk (A.A.A.); dibesh.thapa@kcl.ac.uk (D.T.); anna.starr@gmail.com (A.S.); paul.fraser@kcl.ac.uk (P.A.F.); keith.farrell-dillon@kcl.ac.uk (K.F.-D.); 2School of Cancer and Pharmaceutical Sciences, King’s College London, London SE1 9NH, UK; 3Programa de Pós-Graduação, em Biotecnologia Aplicada à Saúde da Criança e do Adolescente, Instituto de Pesquisa Pelé Pequeno Príncipe, Curitiba 80230-020, PR, Brazil; elizabeth.fernandes@pelepequenoprincipe.org.br; 4School of Bioscience Education, Faculty of Life Sciences & Medicine, King’s College London, London SE1 1UL, UK

**Keywords:** TRPC5, heart rate, cardiac hypertrophy, cardiac remodelling

## Abstract

TRPC5 is a non-selective cation channel that is expressed in cardiomyocytes, but there is a lack of knowledge of its (patho)physiological role in vivo. Here, we examine the role of TRPC5 on cardiac function under basal conditions and during cardiac hypertrophy. Cardiovascular parameters were assessed in wild-type (WT) and global TRPC5 knockout (KO) mice. Despite no difference in blood pressure or activity, heart rate was significantly reduced in TRPC5 KO mice. Echocardiography imaging revealed an increase in stroke volume, but cardiac contractility was unaffected. The reduced heart rate persisted in isolated TRPC5 KO hearts, suggesting changes in basal cardiac pacing. Heart rate was further investigated by evaluating the reflex change following drug-induced pressure changes. The reflex bradycardic response following phenylephrine was greater in TRPC5 KO mice but the tachycardic response to SNP was unchanged, indicating an enhancement in the parasympathetic control of the heart rate. Moreover, the reduction in heart rate to carbachol was greater in isolated TRPC5 KO hearts. To evaluate the role of TRPC5 in cardiac pathology, mice were subjected to abdominal aortic banding (AAB). An exaggerated cardiac hypertrophy response to AAB was observed in TRPC5 KO mice, with an increased expression of hypertrophy markers, fibrosis, reactive oxygen species, and angiogenesis. This study provides novel evidence for a direct effect of TRPC5 on cardiac function. We propose that (1) TRPC5 is required for maintaining heart rate by regulating basal cardiac pacing and in response to pressure lowering, and (2) TRPC5 protects against pathological cardiac hypertrophy.

## 1. Introduction

The transient receptor potential (TRP) superfamily comprises non-selective cation channels that are polymodal sensors and are important regulators of cellular functions [1]. TRPC5 is a member of the canonical family subgroup (TRPC) of which there are seven known members, six of which are expressed in humans [2]. Functional channels are formed as homotetramers, or through heterotetrameric assembly with other members of the TRPC family, most notably TRPC1 and TRPC4 [3,4]. TRPC5 was originally identified in the central nervous system, but is also known to be located in peripheral tissues, including those of the cardiovascular system [5]. mRNA transcripts and protein have been detected in endothelial cells [6] and vascular myocytes [7,8], as well as in cardiomyocytes [5,9,10]. However, our understanding of the functional role of TRPC5 in the cardiovascular system is at an early stage.

Whilst roles for other TRP channels in the cardiovascular system are established, TRPC5 has not been as intensely studied, partly due to a lack of availability of selective ligands. An early report stated that endothelin-1 activates store-operated currents formed from TRPC1-TRPC5-TRPC6 in coronary artery myocytes [11], whilst the non-selective TRPC4 and TRPC5 agonist Englerin A increases mean arterial pressure in anaesthetised rats [12]. TRPC5 has been proposed to have mechanosensing properties and is able to sense increases in pressure from aortic and carotid baroreceptors in rodents [13]. However, these findings are controversial due to the authors’ subsequent re-analysis of their data, leading to different conclusions, and later stating the inclusion of the non-physiological TRPC5 enhancers lanthanum in their single channel recordings [14]. Global TRPC5 KO mice have unchanged basal blood pressure parameters [13,14,15]. To date, few studies of TRPC5 in the heart have been performed. The non-selective TRPC antagonist SKF-96365 produced negative chronotropic, ionotropic, and dromotropic effects in embryo chick hearts [16] and comparable results have been observed with the compound in mice, but TRPC5 was not detected in the sinoatrial node [17].

Little is known about TRPC5 in cardiovascular pathology beyond expression studies. In vitro silencing of SERCA2a in cultured rat cardiomyocytes resulted in an increased TRPC5 expression [18], implicating a potential role in cardiac disease. In rodents, an increased expression has been noted in failing cardiac left ventricles of spontaneously hypertensive rats [19], but studies using a pressure overload model of cardiac hypertrophy have implicated TRPC1 [10], TRPC3 [9,20], and TRPC6 [21] in pathogenesis, whilst TRPC5’s expression was reported to be unchanged [9,10]. In humans, its expression was increased in left ventricle biopsies from patients with end-stage dilated cardiomyopathy [9], suggesting that TRPC5 may be more relevant in humans or severe cardiac dysfunction. A report described a protective role for TRPC5 in the ATP-induced hypertrophy of cultured rat neonatal cardiomyocytes [22], providing the first functional evidence to implicate this channel in cardiac pathology. Critically, however, there is no direct study that has implicated TRPC5 in the pathogenesis of cardiac pathology in vivo.

In this study, we demonstrate that global TRPC5 KO mice present with a reduced heart rate phenotype, which appears to be both driven by alterations in cardiac pacing and an enhanced cardiac response to the parasympathetic drive. Furthermore, our results reveal that TRPC5 has a protective phenotype against pressure overload-induced cardiac hypertrophy and fibrosis. To our knowledge this is the first report that implicates TRPC5 in cardiac function.

## 2. Methods and Materials

### 2.1. Animals

The UK Home Office Project Licence number is 7007959 held under the Animal Scientific Procedures Act (1986). The licence was granted on 19 December 2016 and approved by the King's College London Animal Care and Ethics Committee on 7007959. In vivo experimental design complied with Animal Research: Reporting of In Vivo Experiments (ARRIVE) guidelines [23]. Animals were housed in a climatically controlled environment (22 ± 2 °C) under a 12 h light (7 a.m.–7 p.m.)/dark (7 p.m.–7 a.m.) cycle, with free access to chow and water ad libitum. WT and global TRPC5 KO mice were bred in-house, where male age-matched littermates (8–12 weeks, 20–30 g) were used for investigation purposes. WT and TRPC5 KO mice were kindly gifted by Professor David E. Clapham (Harvard Medical School, Boston, MA, USA) and generated as previously described [24]. Mice were randomly allocated to procedures and the investigators were blinded where appropriate. Following the end of experiments, all mice were euthanised by cervical dislocation, unless otherwise specified.

### 2.2. Radio Telemetry of Conscious Haemodynamics

Baseline haemodynamics were assessed in animals using PA-C10 radio-telemetry devices (Data Science International, New Brighton, MN, USA), as previously described [25]. Briefly, mice were anaesthetised with 1.5–2% isoflurane (Isocare, Abbott Laboratories, Surrey, UK) carried in 95% O_2_/5% CO_2_ (The BOC Group, London, UK) and preoperative analgesia was provided (50 µg/kg buprenorphine, i.m., Vetergesic, Alstoe Animal Health, Yorkshire, UK). Under aseptic conditions, the catheter of the telemetry device was surgically inserted into the left common carotid artery and advanced towards the aortic arch before being secured with 5-0 non-absorbable silk sutures threads (Mersilk, Ethicon, San Angelo, TX, USA). The body of the telemetry device was placed in an s.c. pocket in the right flank. Mice were allowed to recover for two weeks before recording baseline blood pressures, heart rate, and activity. Basal haemodynamic parameters of mice were measured across a 2 light/3 dark phase period, where parameters were measured for 2 min at 10 min intervals.

Time-domain heart variability analysis was performed on baseline data using the blood pressure pulse interval trace, where beat-to-beat intervals were marked using peak systolic measurements and time between intervals were analysed. Only sinus heart beats were selected, and ectopic beats and artefacts were determined manually and excluded from analysis using the heart rate variability module within LabChart 8 Pro software (ADInstruments, Oxford, UK). Data were separated into light and dark phases and the following measurements were analysed: pulse interval, standard deviation (SD) of the pulse interval, square root mean of squared successive differences between adjacent intervals (rMSSD), and percentage of consecutive interval differences exceeding x msec (pNN*_x_*). Intervals of 10, 20, and 30 msec were chosen based on previous publications [26,27].

### 2.3. Two-Dimensional Ultrasound Echocardiography

Ultrasound echocardiography imaging was performed in accordance with the current guidelines for measuring cardiac physiology in mice [28]. Left ventricle myocardial dimensions and function were assessed in anaesthetised (1.5–2% isoflurane carried in 95% O_2_/5% CO_2_) naïve mice and those subjected to abdominal aortic banding. Left ventricle images were taken using a Vevo 770 Ultrasound Imaging System with a 30 MHz linear probe (FujiFilm VisualSonics, Toronto, ON, Canada). A parasternal long axis view of the left ventricle was taken where the apex of the heart was aligned with the root of the aorta and the papillary muscle was in view. M-mode images were taken and analysed at the level of the papillary muscle. Short axis view of the left ventricle was visualised where M-mode images were taken at the level of the papillary muscle.

### 2.4. Haemodynamic Response to Vasoactive Agents

Blood pressure and heart rate changes were determined in anaesthetised mice (1.5–2% isoflurane carried in 95% O_2_/5% CO_2_) following administration of vasoactive compounds. A fluid (saline) cannula (1Fr to 3Fr, model C10PU-MJV1403, Instech Laboratory, Solana Beach, CA, USA) was inserted into the left external jugular vein and secured using 5-0 non-absorbable silk sutures threads. The opposite end of the catheter was connected to a PinPort^TM^ (Instech Laboratory, Solana Beach, CA, USA) to allow i.v. administration of vasoactive compounds. Following this, a separate fluid-filled (100 U/mL heparin in saline) catheter (2 Fr, Smiths Medical-Portex, Hythe, UK) attached to a pressure transducer (ADInstruments, Oxford, UK) was inserted into the left common carotid artery. Following a 15 min stabilisation period, the blood pressure pulse interval was recorded using LabChart 8 Pro software. Phenylephrine (10 µg/kg and 50 µg/kg, dissolved in saline) and SNP (10 µg/kg and 30 µg/kg, dissolved in saline) were administered i.v. through the jugular catheter and the haemodynamic changes were assessed. Mean arterial pressure and heart rate were obtained and averaged to 0.5 s intervals for analysis. Peak changes in mean arterial pressure and heart rate were expressed as change from baseline where the averaged 30 s prior to drug administration was used as baseline. Evaluation of baroreflex sensitivity was assessed in two ways; firstly by determining the ratio of change in heart rate to change in pressure, and secondly by calculating the rate of heart rate change against the pressure change through linear regression analysis as described in [29,30]. Low- and high-dose data were combined and the bradycardic (phenylephrine) or tachycardic (SNP) response was plotted against the corresponding change in pressure. A linear regression was plotted for individual responses and the averaged slope of the regression line was used as a determinant of baroreflex sensitivity.

### 2.5. Langendorff Isolated Perfused Heart

Mice were terminally anaesthetised using pentobarbital (300 mg/kg in saline, i.p., Merial, London, UK) containing 150 units heparin. Hearts were excised and set up on a Langendorff apparatus by cannulating the aorta with blunted 23G needle. Hearts were perfused at a constant pressure of 80 mmHg through the aorta with a modified Krebs–Henseleit solution (118.5 mM NaCl, 25 mM NaHCO_3_, 1.2 mM MgSO_4_, 4.7 mM KCl, 1.2 mM KH_2_PO_4_, 11 mM D-glucose, and 1.4 mM CaCl_2_, all salts from Sigma Aldrich, Dorset, UK) gassed with 95% O_2_/5% CO_2_ and maintained at 37 °C. Following equilibration, hearts were perfused with isoprenaline (1 nM followed by 10 nM) or carbachol (1 nM followed by 10 nM) and heart rate was recorded.

### 2.6. Suprarenal Abdominal Aortic Banding (AAB) Model of Cardiac Hypertrophy

Mice were subjected to pressure overload-induced cardiac hypertrophy as previously described [25]. Briefly, mice were anaesthetised (1.5–2% isoflurane carried in 95% O_2_/5% CO_2_) and preoperative analgesia was provided (50 µg/kg buprenorphine, i.m.). Under aseptic conditions, the abdominal aorta was partially ligated to the width of a 28G needle (0.36 mm) in between the celiac artery and superior mesenteric artery bifurcations with an 8-0 non-absorbable polyamide suture thread (Ethilon, Ethicon, Cincinnati, OH, USA). Sham surgeries were performed in a similar manner with the notable exclusion of the ligation. Cardiac hypertrophy was allowed to develop for either 4 or 10 weeks, after which cardiac function was assessed through 2-dimensional ultrasound echocardiography and organs were harvested for further investigations.

### 2.7. Histology

Cardiac left ventricles were flushed and fixed in 4% paraformaldehyde (Sigma-Aldrich, Dorset, UK) before being embedded in paraffin wax. Tissue was sectioned transversely at a thickness of 6 µm using a microtome (HM 330, MicroM, Heidelberg, Germany) onto poly-lysine slides (Thermo Scientific, Kent, UK).

Interstitial and perivascular fibrosis were assessed using Picro-Sirius Red (0.1% *w*/*v* Direct Red 80 in picric acid, Sigma-Aldrich, Dorset, UK). Sections were visualised at ×20 magnification to determine interstitial fibrosis and ×40 magnification for perivascular fibrosis using Lietz Diaplan microscope (Leica, London, UK) and ProgRes C5 CCD colour camera (Jenoptik, Jena, Germany). Images were captured using ProgRes CapturePro software (version 2.8.8, Jenoptik, Germany) under bright field and circular polarised light. Circular polarised light images were analysed using ImageJ software (version 1.48, NIH, Bethesda, MD, USA). Eight images per section and six sections per heart were analysed to determine interstitial fibrosis where data were expressed as percentage area of field of view. All blood vessels were imaged per section; six sections per heart were analysed for perivascular fibrosis where cross-sectional surface area was normalised to total vessel wall cross-sectional area.

Rhodamine conjugated wheat germ agglutinin (WGA, 1-in-200 dilution RL-1022, Vector Laboratories, Plain City, OH, USA) was used to stain the plasma membrane of cardiomyocytes. Sections were visualised at ×40 magnification using a Nikon Diaphot 300 microscope (Nikon, Tokyo, Japan) and ORCA-03G CCD black and white digital camera (Hamamatsu, Japan) and captured using Image Hopper software (version 2.0, Samsara Research, Oxford, UK). Cardiomyocyte cross-sectional surface area was determined from eight images per section, four sections per heart were analysed using ImageJ software (version 1.48), where area was determined for 20 random cardiomyocytes per section.

Endothelial cells were stained with biotinylated Isolectin B_4_ (IB_4_, 1-in-100 dilution, B-1205, Vector Laboratories) and DyLight 488 streptavidin (1-in-200 dilution SA-5488, Vector Laboratories), to outline capillaries. Sections were also counter-stained with rhodamine conjugated WGA. Sections were visualised at ×40 magnification, and total capillaries and cardiomyocytes were counted per image. Eight images per section, four sections per heart were analysed using ImageJ software (version 1.48). Data were expressed as capillaries per cardiomyocytes. Colour image overlays were created in Adobe Photoshop CC (version 14, Adobe Systems Inc., San Jose, CA, USA).

### 2.8. RT-qPCR

Total RNA was extracted from either whole hearts or left ventricles, and whole brains using the Qiagen Microarray RNA extraction kit (Qiagen, Manchester, UK), according to the manufacturer’s guidelines. An amount of 1 µg of total RNA was converted into cDNA using the High Capacity RNA-to-cDNA kit (Applied Biosystems, Middletown, CT, USA) according to the manufacturer’s instruction. Negative RT mixtures were also aliquoted (reaction mixtures without RT enzyme) to provide a negative control, after which samples were diluted 1:40 (1:10 in mesenteric arteries). qPCR was conducted using SybrGreen-based mix (Brilliant III Ultra-Fast SYBRGreen QPCR Master Mix, Agilent, Didcot, UK), and primers specific for the gene of interest. Reaction mixtures were pipetted using the RCorbett CAS-1200 automated PCR setup (Qiagen, UK), and PCR using the Rotor-Gene 6000 thermal cycler (Qiagen, UK) using the following conditions: initial denaturation 95 °C for 10 min; 45 cycles of 95 °C for 10 s, 57 °C for 15 s, and 72 °C for 5 s; melt 68–95 °C. A melt curve was generated to determine number of products created and primer specificity. Sample concentrations were calculated using the given standard concentration using Rotor-Gene Q Series software (version 2.3.1, Qiagen, UK). Data were expressed as copies/µL and normalised to *Actb* (β-actin), *B2m* (β-2-microglobulin), and *Hprt* (hypoxanthine guanine phosphoribosyl transferase) using GeNorm v3.4 software. Primers are listed in Appendix A.

### 2.9. Immunoblotting

Cardiac left ventricles were homogenised using an SDS-based lysis buffer containing protease inhibitor cocktail (1 tablet per 10 mL lysis buffer, Roche, Herts, UK), and protein concentration determined using the colorimetric Lowry copper based protein assay protocol [31]. Samples were loaded onto a 12% Tris-SDS-polyacrylamide gel for protein separation according to size, after which proteins were transferred onto a PVDF membrane. The membrane was treated with 5% *w*/*v* BSA for 1 h at room temperature to block unbound pores. Membranes were incubated with primary antibody, either at room temperature for 1 h or overnight at 4 °C, and species-specific horseradish peroxidase-conjugated secondary antibodies at room temperature for 1 h. Chemiluminescence was detected by incubating membranes with 1 mL enhanced chemiluminescence substrate (Merck Millipore, London, UK) for 1 min before developing membranes in the G-Box gel documentation system (G-Box, Syngene, Cambridge, UK) and images captured using GeneSnap software (version 7.12, Syngene, UK). Primary antibodies against gp91(phox) (611414, BD Biosciences, Berks, UK), HO-1 (ab13243, Abcam, Cambridge, UK), nitrotyrosine (ab61392, Abcam, UK), and GAPDH (AM4300, Ambion, Oxford, UK) were used in this study. Densitometry analysis was performed using ImageJ software (version 1.48), and normalised to the loading control GAPDH.

### 2.10. Statistical Analysis

Investigators were blinded to the study groups where appropriate. All data were placed in Microsoft Excel 2007 (Microsoft Corporation, Redmond, WA, USA) and analysed in GraphPad Prism software (version 9.4.1, GraphPad Software Inc., Solana Beach, CA, USA). All values are illustrated as mean ± standard error of the mean (s.e.m); *n* represents the number of animals. Data containing two groups were analysed using Student’s *t*-test, whilst data containing multiple groups were analysed using two-way analysis of variance (ANOVA) followed by the Bonferroni post-hoc correction. *p* < 0.05 was regarded as statistically significant.

## 3. Results

### 3.1. Baseline Reduction in Basal Heart Rate in Global TRPC5 KO Mice

To compare baseline cardiovascular heamodynmics, we surgically implanted radiotelemetry devices into WT and global TRPC5 KO mice. Systolic, diastolic, and mean arterial pressure were comparable between genotypes (Appendix A and Figure 1A). Heart rate was significantly reduced in TRPC5 KO mice (Figure 1B), independent of changes in activity (Figure 1C). Cardiac mass and left ventricular wall thicknesses were comparable between genotypes (Appendix A). Left ventricular lumen diameter and volume were increased in TRPC5 KO mice only during diastole (Lumen diameter: WT 3.69 ± 0.13 mm vs. TRPC5 KO 3.98 ± 0.06 mm, (*n* = 8–14, *p* < 0.05); volume: WT 58.39 ± 4.47 vs. TRPC5 KO 69.52 ± 2.35 (*n* = 8–14, *p* < 0.05)). Consequently, stroke volume was elevated in TRPC5 KO mice (Figure 1D), whilst cardiac output was comparable to their WT counterparts (Figure 1E). Ejection fraction and fractional shortening, markers of cardiac contractility, were comparable between genotypes (Figure 1F,G).

Further analysis of heart rate was performed using the time-domain heart rate variability of the pulse interval to provide an indication of autonomic regulation. The pulse interval was significantly increased in TRPC5 KO mice (Figure 1H), as was the standard deviation of the pulse interval (Figure 1I). A trend towards an increase in the square root mean of squared successive differences between adjacent intervals (rMSSD) in the TRPC5 KO cohort was observed (Figure 1J), and the percentage of consecutive interval differences (pNN) exceeding 10, 20, and 30 ms was significantly increased (Figure 1K–M), suggesting a possible enhancement in cardiac parasympathetic innervation in TRPC5 KO mice.

It is possible that changes in other TRPC subunits could compensate for the reduced expression of TRPC5 in global TRPC5 KO animals. Consequently, mRNA levels of canonical subunits were determined but no difference was observed in the expression of other TRPC channels in the heart (Appendix A) or brain (Appendix A).

### 3.2. Cardiac Pacing Abnormalities and Enhanced Cardiac Response to Cholinergic Stimulation in TRPC5 KO Mice

TRPC5 has been suggested to be involved in mediating the baroreflex response following acute pressure increases [13,14]. To assess this in our mice, we examined changes in blood pressure and heart following the acute administration of phenylephrine and SNP. Phenylephrine (10 and 50 µg/kg) produced an increase in blood pressure that was comparable in WT and TRPC5 KO mice (Figure 2A–C). However, the reflex bradycardic response was enhanced in the TRPC5 KO cohort (Figure 2D). Analysing the change in heart rate to mean arterial pressure ratio (Figure 2E) and linear regression of the rate of heart rate reduction (Figure 2F) further confirmed the enhanced sensitivity following pressure increase. The reflex response to SNP was comparable between genotypes (Appendix A). Taken together these data suggest an enhanced bradycardic response to pressure increase in TRPC5 KO mice, whilst the tachycardic response is unaffected.

To further assess heart rate responses between WT and KO mice, we employed the isolated heart Langendorff model. The spontaneous heart rate of TRPC5 KO hearts were found to be lower compared to WT (Figure 2G), suggesting that pacing abnormalities may contribute to the reduced heart rate phenotype. To elucidate further, the hearts were perfused with isoprenaline to increase heart rate, but no differences were observed between genotypes (Figure 2H), consistent with the unchanged reflex tachycardic response to SNP. However, heart rate reduction in response to the muscarinic receptor agonist carbachol was more pronounced in the TRPC5 KO cohort (Figure 2I), further implicating TRPC5 in the control of heart rate.

### 3.3. TRPC5 Protects against Pressure Overload-Induced Cardiac Hypertrophy

We assessed the role of TRPC5 in cardiac hypertrophy by subjecting mice to aortic pressure overload through suprarenal aortic banding. After 4 weeks, heart weight was significantly increased in both WT and TRPC5 KO mice, but the increase was more pronounced in the TRPC5 KO cohort (Figure 3A). The enhanced elevation in heart weight in TRPC5 KO mice was retained, but not further increased when the banding duration was increased to 10 weeks. The increase in heart weight was reflected by an increase in left ventricular mass, indicating the presence of left ventricular cardiac hypertrophy, with an exacerbated phenotype observed in TRPC5 KO mice (Figure 3B). Left ventricular dimensions were further assessed using ultrasound echocardiography. The posterior wall thickness was increased in banded mice, indicative of concentric hypertrophy, with an exaggerated response in the TRPC5 KO cohort (Figure 3D,E). Banding did not affect the lumen diameter in either WT or TRPC5 KO mouse, suggesting that the pathology did not reach the level of ventricular dilatation (Appendix A). Transverse sections of the left ventricle were stained with WGA to visualise the cell membrane. An exaggerated increase in cardiomyocyte cross-sectional surface area was observed in banded TRPC5 KO mice (Figure 3F–H), indicative of cardiomyocyte expansion. Both ejection fraction and fractional shortening were unchanged, despite extending the banding duration to 10 weeks (Figure 3I,J).

Studies have implicated other canonical subunits in the pathogenesis of cardiac hypertrophy. Thus, it was important to investigate the TRPC mRNA expression profile in the left ventricle. Expression levels of other TRPC subunits were unchanged in the present study (Appendix A) but, notably, TRPC5 transcripts were found to be reduced in banded WT mice (Figure 4A).

The expression profile of markers associated with cardiac hypertrophy were also assessed in left ventricles. The atrial natriuretic peptide (ANP) expression was significantly increased by 10 weeks, with a further elevation observed in left ventricles of banded TRPC5 KO mice (Figure 4B). The sarcomeric proteins β-myosin heavy chain (β-MHC) (Figure 4C) and skeletal α-actin (Figure 4D) were also significantly elevated in left ventricles of banded TRPC5 KO mice compared to WT controls. The exacerbated phenotype observed in TRPC5 KO mice was accompanied by increased reactive oxygen species (ROS) generation as observed through the increased protein expression of gp91(phox) (Figure 4E, Appendix A), and nitrosative stress, as shown by increased nitrotyrosine levels (Figure 4F, Appendix A). Interestingly, a time-dependent change was observed in the antioxidant defence protein heme oxygenase-1 (HO-1); HO-1 protein was elevated at 4 weeks in TRPC5 KO left ventricles but not at 10 weeks, whilst it was only increased in WT at the 10-week period (Figure 4G, Appendix A). This expression pattern was also mirrored by the transforming growth factor-β_1_ (TGF-β_1_) mRNA (Figure 4H). Combined, these data suggest that the exacerbated hypertrophy response in left ventricles of TRPC5 KO was accompanied by an increase in expression hypertrophy-related genes.

### 3.4. Exaggerated Hypertrophy Is Accompanied by Increased Left Ventricular Remodelling

We assessed cardiac remodelling in hypertrophied hearts by assessing collagen deposition and capillary rarefaction. Increased expression of collagen type I α1 mRNA was observed in left ventricles of banded TRPC5 KO mice, but not WT at both time points (Figure 5A), whilst an increase in collagen type I α2 was seen at the 10-week time point (Figure 5B). This was also reflected in the Picro-Sirius Red histological assessment as both interstitial (Figure 5C,D,G) and perivascular fibrosis (Figure 5E,F,H) were only present in left ventricles of banded TRPC5 KO mice. Capillary density was significantly increased in left ventricles of banded TRPC5 KO mice, compared to WT at both timepoints (Figure 6), in keeping with a hypertrophic phenotype.

## 4. Discussion

Our results demonstrate that TRPC5 deletion leads to a reduced basal heart rate and a consequential increase in stroke volume in order to maintain cardiac output. We propose that this is due to changes in cardiac pacing and an enhanced response to cholinergic stimulation. The results further reveal that TRPC5 has a protective phenotype against maladaptive cardiac hypertrophy. TRPC5 KO mice subjected to aortic pressure overload presented with an exacerbated hypertrophy phenotype, increased oxidative stress, fibrosis, and angiogenesis. This is a novel demonstration of an in vivo role, which is independent of expression changes in the other TRPC channels in the heart.

We observed similar systemic blood pressure readings in WT and TRPC5 KO mice, consistent with previous reports [13,14]. Despite no change in blood pressure, a reduced heart rate was detected in TRPC5 KO mice. The lower heart rate, with an increased duration of cardiac diastole, resulted in increased stroke volume, but cardiac output was comparable to WT, with no change in cardiac contractility. Isolated heart investigations using the Langendorff preparation revealed that the reduced heart rate persisted in TRPC5 KO hearts ex vivo, suggesting that TRPC5 may regulate cardiac pacing. The non-selective TRPC agonist SKF-96365 has been shown to reduce heart rate [16,17]. However, TRPC5 mRNA transcripts were not detected in the sinoatrial node [17]. This suggests that the reduced heart rate may be explained by reduced transduction through the atrioventricular node, rather than changes in the pacemaker potential.

Both rMSSD and pNN*_x_* have been suggested to reflect changes in variations due to parasympathetic modulation in heart rate variability analysis [26,27,32,33]. Whilst a modest increase in rMSSD was observed in TRPC5 KO mice, pNN*_x_* was significantly increased suggesting a possible increase in the parasympathetic modulation of heart rate in TRPC5 KO mice. This was confirmed through the assessment of cardiac reflex control. The bradycardic response following phenylephrine administration was potentiated in TRPC5 KO animals, whilst the tachycardic response to SNP was unaffected. This suggests that, whilst the parasympathetic arm of the baroreflex cascade is sensitised in TRPC5 KO mice, the sympathetic arm is unchanged. Lau et al. (2016) have reported an attenuated bradycardic baroreceptor response in global TRPC5 KO animals and proposed a role for TRPC5 in afferent nerve mechanosensation within the aortic and carotid baroreceptor regions [13]. Conversely, our results point to an enhanced bradycardic response in TRPC5 KO mice, due to changes in the efferent arm of the reflex cascade. The reasons for these discrepancies are not clear, but investigations in isolated TRPC5 KO hearts further implicate the channels in parasympathetic signalling. The muscarinic receptor agonist carbachol produced a greater reduction in heart rate, suggesting that the enhanced parasympathetic response occurs at the level of the cardiomyocyte and not from changes in neuronal signalling. It is well established that cardiac M_2_ receptors are responsible for reducing heart rate following parasympathetic signalling [34,35]. Previous reports have linked M_2_ receptor signalling to TRPC5 activity in HEK cells [36], but further investigations are required to determine whether this occurs in cardiomyocytes.

Cardiac pathology was investigated using suprarenal AAB to induce cardiac hypertrophy. Aortic banding produced an increase in cardiac left ventricular mass that was further elevated in TRPC5 KO mice. Ultrasound echocardiography confirmed the presence of hypertrophy due to an increase in posterior wall thickness. The internal dimensions, however, were not altered, suggesting that hypertrophy was not accompanied by ventricular dilatation. The analysis of the cross-sectional surface area indicated that the increase in mass was due to the increased expansion of cardiomyocytes. In contrast to our expectations, extending the banding duration to 10 weeks did not further affect the pathological response to pressure overload, or affect cardiac contractility in either genotype. This is most likely due to the background strain of mice used in this study. A prior study demonstrated that 129S1/SvlmJ mice have a reduced pathological response compared to C57BL/J6 mice following transverse aortic constriction [37].

The exaggerated maladaptive hypertrophy phenotype in TRPC5 KO was further supported by the increased expression profile of cardiac hypertrophy biomarkers. The mRNA expression of the natriuretic peptide ANP, the sarcomeric proteins β-MHC, and skeletal α-actin were significantly enhanced in the left ventricles of TRPC5 KO mice. Increased markers of ROS were noted in the TRPC5 KO cohort, as observed through increased gp91(phox) protein expression and protein nitrosylation. The expression of the antioxidant defence protein HO-1 was increased at 4 weeks in left ventricles of banded TRPC5 KO mice but returned to baseline at 10 weeks, in keeping with the proposal that it plays a role in the early stages of the disease [38]. Whilst HO-1 is known to have cardioprotective effects [39], studies have demonstrated an aggravated hypertrophy phenotype in HO-1 overexpressed transgenic mice [40,41]. The mRNA expression profile of the pleiotropic cytokine TGF-β_1_, known to regulate HO-1 expression [42,43], mimicked that of HO-1. TGF-β_1_ expression has been reported to increase during the early phases of left ventricular hypertrophy in aortic banded rats, but to decline in later stages [44]. TRPC5 channel activity is known to be modulated by ROS and contains S-nitrosylation sites [45], but the significance of this for hypertrophy signalling is not clear.

The heightened pathological response in the TRPC5 KO cohort was accompanied by extensive cardiac remodelling. The increased mRNA expression of collagen type I α chains was detected in left ventricles of banded TRPC5 KO mice only, and histological analysis revealed interstitial and perivascular fibrosis. TRPC5 has been detected in rat cardiac fibroblasts in vitro [46], where it was proposed to contribute towards fibroblast motility and migration [47]. With respect to the present study, the increased fibrosis in the KO cohort is likely due to the exacerbated response to banding, rather than direct effects of TRPC5 in cardiac fibroblasts, albeit further investigations are required to confirm this. Increased capillary density was found in KO banded hearts, in keeping with the overexpression of pro-angiogenic factors increasing cardiac mass [48,49,50].

There is evidence that other canonical subunits are involved in cardiac hypertrophy, most notably Ca^2+^ influx through TRPC1 [10], TRPC3 [9,20], and TRPC6 [21] channels initiates the calcineurin-NFAT hypertrophy signalling pathway. However, no changes in mRNA expression levels of other canonical channels were found in either the WT or TRPC5 KO cohort. Conversely, we did detect a reduction in TRPC5 mRNA in hypertrophic ventricular tissue, in keeping with studies from our laboratory demonstrating the downregulation of TRPC5 is associated with disease [51]. A study using cultured neonatal rat cardiomyocytes proposed a protective role for TRPC5 against ATP-induced cardiac hypertrophy [22]. A TRPC5 and endothelial nitric oxide synthase interaction was required to produce nitric oxide and negatively regulate cardiac hypertrophy signalling [22]. Although these observations were not validated in vivo, the antihypertrophic properties of nitric oxide are widely known [52] and this pathway may explain the protective role for TRPC5 observed in the present study.

## 5. Conclusions

In conclusion, this is the first study directly implicating TRPC5 in cardiac (patho)physiology. We provided novel evidence to support a role for TRPC5 in the regulation of basal heart rate, through altered pacing and an enhanced response to parasympathetic signalling. We further proposed that TRPC5 is able to protect against pathological cardiac hypertrophy. The genetic deletion of TRPC5 resulted in an exaggerated phenotype as mice presented with increased cardiac mass that was accompanied by alteration in gene expression, ROS generation, and cardiac remodelling. These findings reveal the role of TRPC5 as regulatory factor in physiology and pathology, and a potential novel therapeutic target.

## Figures and Tables

**Figure 1 biomolecules-14-00442-f001:**
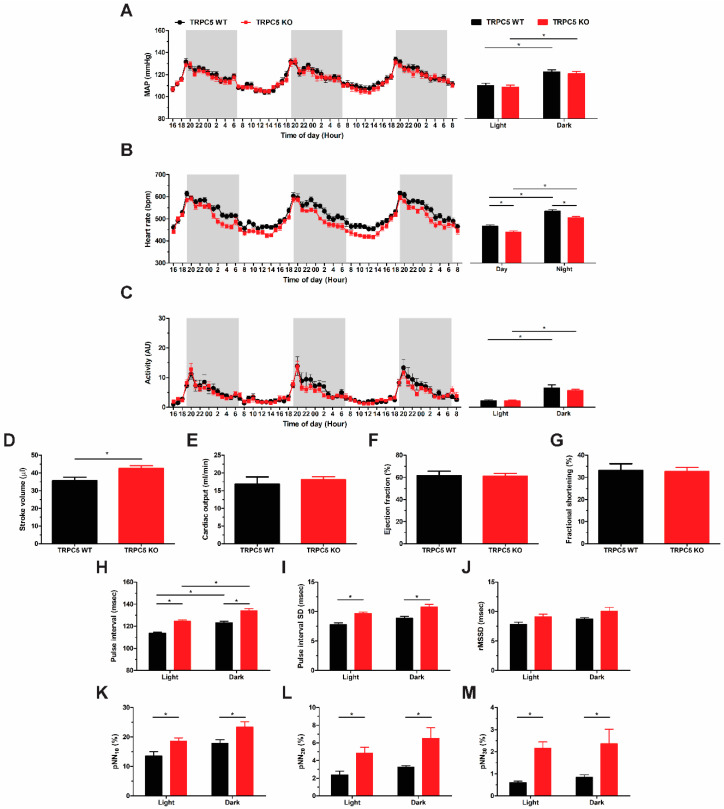
TRPC5 KO mice display a reduced heart rate phenotype. (**A**) Mean arterial pressure (MAP), (**B**) heart rate, and (**C**) activity recordings were obtained from radio-telemetered mice over a 2 light/3 dark period, and shaded regions depict dark phase recordings and bar charts illustrate the average for the light and dark periods (*n* = 19–20). (**D**) Stroke volume, (**E**) cardiac output, (**F**) ejection fraction, and (**G**) fractional shortening were assessed via 2-dimensional echocardiography (*n* = 8–10). Time-domain heart rate variability analysis was performed on the blood pressure pulse waveform and (**H**) pulse interval, (**I**) standard deviation (SD) of the pulse interval, (**J**) square root of the mean of squared successive differences between adjacent intervals (rMSSD), and the percentage of consecutive interval differences (pNN) exceeding (**K**) 10 msec, (**L**) 20 msec, and (**M**) 30 msec were determined and separated into light and dark phases (*n* = 7). Data shown as mean ± s.e.m. * *p* < 0.05 as determined by two-ANOVA followed by Bonferroni post-hoc correction or unpaired Student *t*-test.

**Figure 2 biomolecules-14-00442-f002:**
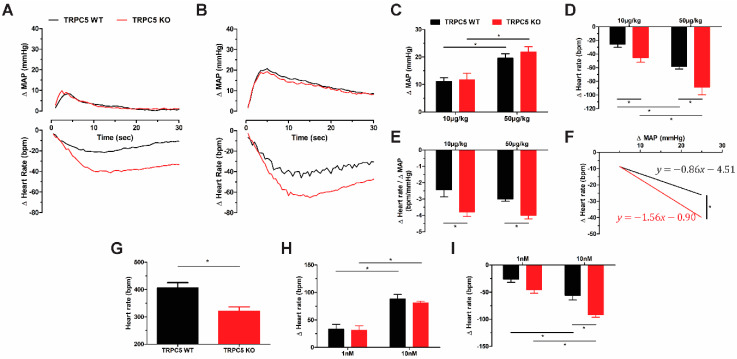
Enhanced cardiac parasympathetic response in TRPC5 KO mice. Changes in mean arterial pressure (MAP) and heart rate were evaluated following phenylephrine i.v. (*n* = 6–8). Black shows responses from TRPC5WT mice and red from TRPC5KO mice. Representative traces depicting changes in mean arterial pressure (MAP) and heart rate following (**A**) 10 µg/kg and (**B**) 50 µg/kg phenylephrine. Peak changes in (**C**) MAP and (**D**) heart rate were assessed, and (**E**) ratio of heart rate over MAP was used a measure of overall baroreflex sensitivity. (**F**) Linear regression for the bradycardic responses following phenylephrine administration. (**G**) Baseline heart rate of isolated hearts on a Langendorff perfusion setup (*n* = 11). Peak change in heart rate in isolated hearts following perfusion with (**H**) isoprenaline (*n* = 5) and (**I**) carbachol (*n* = 6). Data shown as mean ± s.e.m. * *p* < 0.05 as determined by two-ANOVA followed by Bonferroni post-hoc correction.

**Figure 3 biomolecules-14-00442-f003:**
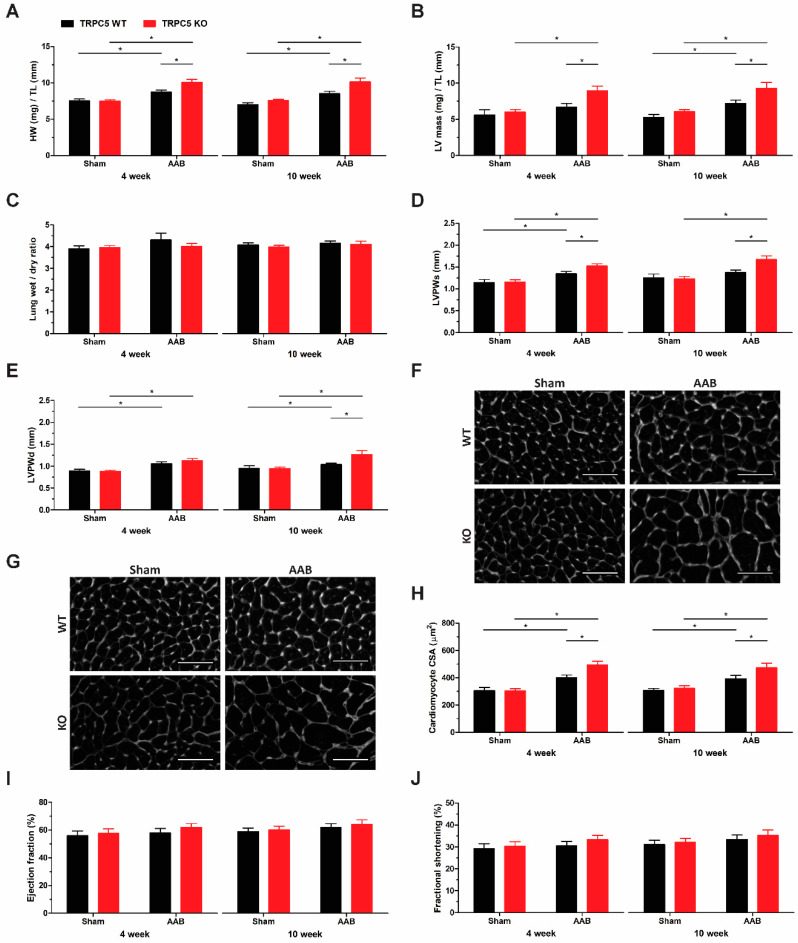
Exacerbated pathological cardiac hypertrophy phenotype in TRPC5 KO mice subjected to pressure overload. WT and TRPC5 KO mice were subjected to either 4 weeks (*n* = 6–9) or 10 weeks (*n* = 10–12) of abdominal aortic banding (AAB) or sham. (**A**) Heart weight and (**B**) left ventricle (LV) mass were normalised to tibia length (TL). (**C**) Lung wet/dry ratio was used to determine presence of pulmonary oedema. LV posterior wall (LVPW) dimensions during (**D**) diastole and (**E**) systole were assessed via 2-dimensional ultrasound echocardiography. Representative images of left ventricular cardiomyocytes stained with the plasma membrane-binding wheat germ agglutinin (WGA) in transverse cardiac sections from mice subjected to (**F**) 4 weeks and (**G**) 10 weeks of pressure overload. Scale bar represents 50 µm (**H**) Cross-sectional surface area (CSA) was calculated for whole cardiomyocytes using ImageJ software (version 1.48). (**I**) Ejection fraction and (**J**) fractional shortening were also determined by echocardiography. Data shown as mean ± s.e.m. * *p* < 0.05 as determined by two-ANOVA followed by Bonferroni post-hoc correction.

**Figure 4 biomolecules-14-00442-f004:**
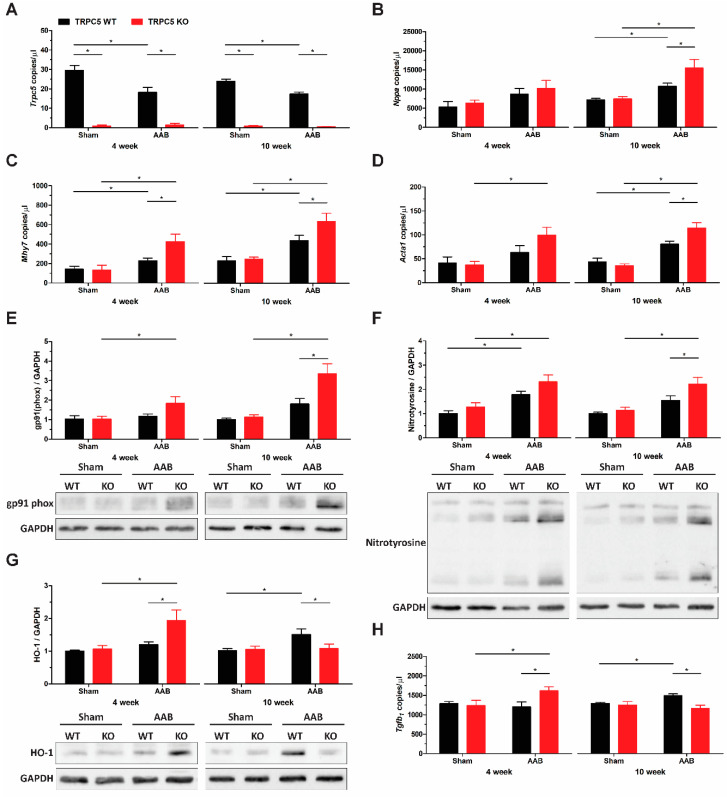
Increased expression of hypertrophy and ROS markers in left ventricles of TRPC5 KO mice following pressure overload. (**A**) *Trpc5*, (**B**) *Nppa* (atrial natriuretic peptide), (**C**) *Mhy7* (β-myosin heavy chain), (**D**) *Acta1* (skeletal α-actin), and (**H**) *Tgfb1* (transforming growth factor-β1) gene expression were assessed via RT-qPCR in cardiac left ventricles from WT and TRPC5 KO mice subjected to 4 weeks (*n* = 5–7) and 10 weeks (*n* = 10–12) of abdominal aortic banding (AAB) or sham. Data were normalised to the reference genes *B2m* (β-2-microglobulin) and *Hprt* (hypoxanthine guanine phosphoribosyl transferase) and expressed as copies/µL. Representative immunoblots and densitometry analysis of (**E**) gp91(phox), (**F**) nitrotyrosine, and (**G**) heme oxygenase 1 (HO-1) protein expression from cardiac left ventricles. Data shown as mean ± s.e.m. * *p* < 0.05 as determined by two-ANOVA followed by Bonferroni post-hoc correction. (**E**–**G**) Original western blots can be found at Appendix A.

**Figure 5 biomolecules-14-00442-f005:**
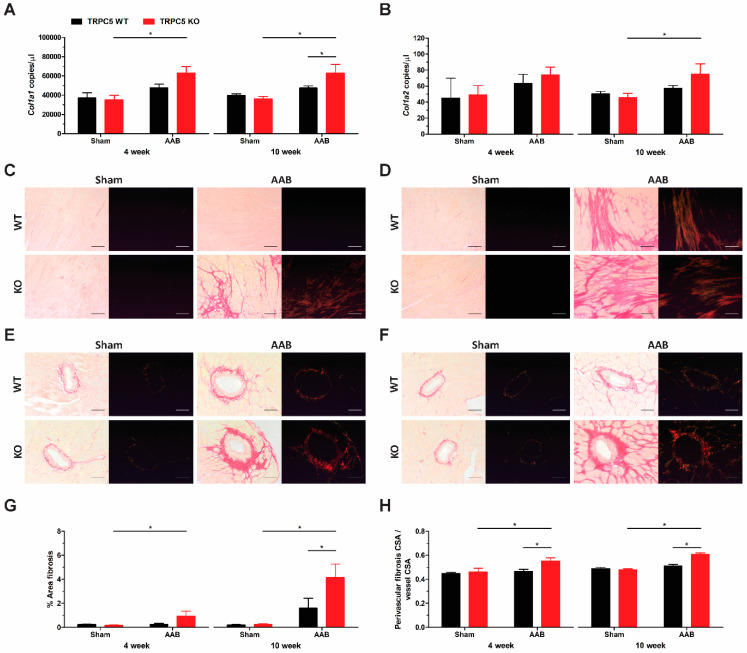
Increased fibrotic remodelling of cardiac left ventricles of TRPC5 mice subjected to pressure overload. (**A**) *Col1a1* (collagen type I α1) and (**B**) *Col1a2* (collagen type I α2) gene expression were assessed via RT-qPCR in cardiac left ventricles from WT and TRPC5 KO mice subjected to 4 weeks (*n* = 5–7) and 10 weeks (*n* = 10–12) of abdominal aortic banding (AAB) or sham. Data were normalised to the reference genes *B2m* (β-2-microglobulin) and *Hprt* (hypoxanthine guanine phosphoribosyl transferase) and expressed as copies/µL. Representative images of (**C**,**D**) interstitial and (**E**,**F**) perivascular fibrosis of (**C**,**E**) for 4-week (*n* = 5–7) and (**D**,**F**) 10-week (*n* = 10–12) transverse cardiac left ventricle sections stained with Picro-Sirius Red, viewed under phase light (left) and circular polarised light (right); bar represents (**C**,**D**) 100 µm and (**E**,**F**) 50 µm. (**G**) Quantification of interstitial fibrosis was determined as percentage per field of view. (**H**) Cross-sectional surface area (CSA) of perivascular fibrosis was calculated and expressed as a ratio of total vessel cross-sectional surface area. All image analysis was performed using ImageJ software (version 1.48). Data shown as mean ± s.e.m. * *p* < 0.05 as determined by two-ANOVA followed by Bonferroni post-hoc correction.

**Figure 6 biomolecules-14-00442-f006:**
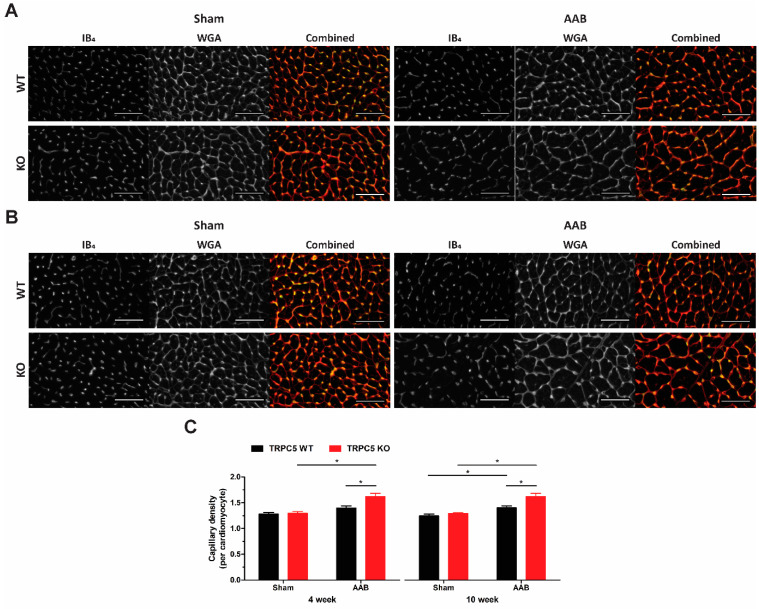
Increased capillary density of cardiac left ventricles of TRPC5 mice subjected to pressure overload. Representative images of capillary density in transverse cardiac left ventricle sections from WT and TRPC5 KO mice subjected to (**A**) 4 weeks (*n* = 5–7) and (**B**) 10 weeks (*n* = 10–12) of abdominal aortic banding (AAB) or sham. Endothelial cells were stained with Isolectin B_4_ (IB_4_), and cardiomyocyte plasma membranes with wheat germ agglutinin (WGA). Colour image overlays were created in Adobe Photoshop CC software; IB_4_ green and WGA red with overlay in yellow; bar represents 50 µm. (**C**) Quantification was determined as total number of capillaries as a ratio to total number of cardiomyocytes per field of view using ImageJ software (version 1.48). Data shown as mean ± s.e.m. * *p* < 0.05 as determined by two-ANOVA followed by Bonferroni post-hoc correction.

## Data Availability

Data is contained within the article and Appendix A.

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
