# Peer review of "Transient Receptor Potential Canonical 5 (TRPC5): Regulation of Heart Rate and Protection against Pathological Cardiac Hypertrophy"

_biomolecules, 2024, doi:10.3390/biom14040442_

Round 1

Reviewer 1 Report

Comments and Suggestions for Authors

This is an interesting, well-crafted manuscript. The aims are clearly set out and the methodology appropriate. In a novel approach, the authors show that TRPC5 knockdown is adversely associated with hypertrophic development and in a reduction in ventricular pacing both in situ and in the isolated Langendorff perfused hearts. The timelines of 4 to 10 weeks used to study the development of ventricular remodelling is entirely appropriate for the aortic banded mouse model. The findings are novel in that they strongly indicate a cardioprotective role for TRPC5 at least in the early stages after the induction of pressure overload cardiomyopathy. In brief, this study has clinical value and merits further examination.

Comments:
Reference is made to the Supplementary tables such as on line 255 in the Result section. I'm afraid that I was unable to find these in the documents submitted to the reviewer. I think however, that it would be worth including this data on diastolic LV lumen diameter and volume, within the main body of Figure 1 or as a separate table within the text.

The authors refer to the controversial findings reported in Correspondence by Thakore et al (Ref 14). It would be useful to expand on this note in the introduction (line 55).

Its a shame that ECG recordings were not taken during echocardiography to confirm AV block. The authors' suggestion that this conduction defect may explain the reduced heart rate, however, stacks up nicely with similar studies. Perhaps an experiment for future studies.

Minor typos:
Line 97: The name of the anaesthetic isoflurane is repeated twice either side of the vaporiser dialled percentage.

Line 101: please correct to read either aortic arch or aorta....... as in, the catheter was advanced towards the aortic arch.

Author Response

We thank the reviewers for their positive feedback and thoughtful comments. We have revised the manuscript according to the suggestions in the revised manuscript and supplementary information. All changes to the manuscript have been noted using the “Track Changes” feature. Our response to reviewer specific comments are below:

Reviewer 1:

This is an interesting, well-crafted manuscript. The aims are clearly set out and the methodology appropriate. In a novel approach, the authors show that TRPC5 knockdown is adversely associated with hypertrophic development and in a reduction in ventricular pacing both in situ and in the isolated Langendorff perfused hearts. The timelines of 4 to 10 weeks used to study the development of ventricular remodelling is entirely appropriate for the aortic banded mouse model. The findings are novel in that they strongly indicate a cardioprotective role for TRPC5 at least in the early stages after the induction of pressure overload cardiomyopathy. In brief, this study has clinical value and merits further examination.

Response: We really appreciate your positive feedback.

Reference is made to the Supplementary tables such as on line 255 in the Result section. I'm afraid that I was unable to find these in the documents submitted to the reviewer. I think however, that it would be worth including this data on diastolic LV lumen diameter and volume, within the main body of Figure 1 or as a separate table within the text.

Response: Apologies, we are unsure what happened to the Supplementary tables and figures. This has been reuploaded with the revised manuscript. In addition to the data table included in the supplemental information, we have also included the values for the LV lumen diameter and volume during diastole in text when describing these data between WT and TRPC5 KO mice.

The authors refer to the controversial findings reported in Correspondence by Thakore et al (Ref 14). It would be useful to expand on this note in the introduction (line 55).

Response: We have expanded on the controversial findings reported in the Correspondence by Thakore et al in the introduction of the manuscript.

Its a shame that ECG recordings were not taken during echocardiography to confirm AV block. The authors' suggestion that this conduction defect may explain the reduced heart rate, however, stacks up nicely with similar studies. Perhaps an experiment for future studies.

Response: We agree that testing for AV block is an interesting avenue for future studies.

Line 97: The name of the anaesthetic isoflurane is repeated twice either side of the vaporiser dialled percentage.

Response: Thank you, we have corrected this repetition.

Line 101: please correct to read either aortic arch or aorta....... as in, the catheter was advanced towards the aortic arch.

Response: Thank you, we have corrected this to read “aortic arch”.

Reviewer 2 Report

Comments and Suggestions for Authors

The manuscript deals with a very specific type of receptor that is involved in heart rate regulation and protection against cardiac hypertrophy. Data presented clearly supposrt the author´s claims and are presented in comprehensible form.

 I have only one minor comment. The authors do not cite M.L. Lindsey et al. "Guidelilnes for measuring cardiac physiology in mice" published in American Journal of Physiology https://doi.org/10.1152/ajpheart.00339.2017 . I consider it worthwhile mentioning in discussion, or maybe methods, in relation to all the methods used in the manuscript. 

Author Response

Review 2:

The manuscript deals with a very specific type of receptor that is involved in heart rate regulation and protection against cardiac hypertrophy. Data presented clearly support the author´s claims and are presented in comprehensible form.

Response: Thank you for your positive feedback.

I have only one minor comment. The authors do not cite M.L. Lindsey et al. "Guidelines for measuring cardiac physiology in mice" published in American Journal of Physiology https://doi.org/10.1152/ajpheart.00339.2017 . I consider it worthwhile mentioning in discussion, or maybe methods, in relation to all the methods used in the manuscript.

Response: We have included the citation in the methods section when describing the procedure for ultrasound echocardiography.